# You Only Propagate Once: Accelerating Adversarial Training via Maximal Principle

**Dinghuai Zhang**[∗]**, Tianyuan Zhang**[∗]
Peking University
{zhangdinghuai, 1600012888}@pku.edu.cn

**Yiping Lu**[∗]
Stanford University
yplu@stanford.edu

**Zhanxing Zhu**[†]
School of Mathematical Sciences, Peking University
Center for Data Science, Peking University
Beijing Institute of Big Data Research
zhanxing.zhu@pku.edu.cn

**Bin Dong**[†]
Beijing International Center for Mathematical Research, Peking University
Center for Data Science, Peking University
Beijing Institute of Big Data Research
dongbin@math.pku.edu.cn

## Abstract

Deep learning achieves state-of-the-art results in many tasks in computer vision and natural language processing. However, recent works have shown that deep networks can be vulnerable to adversarial perturbations, which raised a serious robustness issue of deep networks. Adversarial training, typically formulated as a robust optimization problem, is an effective way of improving the robustness of deep networks. A major drawback of existing adversarial training algorithms is the computational overhead of the generation of adversarial examples, typically far greater than that of the network training. This leads to the unbearable overall computational cost of adversarial training. In this paper, we show that adversarial training can be cast as a discrete time differential game. Through analyzing the Pontryagin's Maximum Principle (PMP) of the problem, we observe that the adversary update is only coupled with the parameters of the first layer of the network. This inspires us to restrict most of the forward and back propagation within the first layer of the network during adversary updates. This effectively reduces the total number of full forward and backward propagation to only one for each group of adversary updates. Therefore, we refer to this algorithm YOPO (**Y**ou **O**nly **P**ropagate **O**nce). Numerical experiments demonstrate that YOPO can achieve comparable defense accuracy with **approximately 1/5 ∼ 1/4 GPU time** of the projected gradient descent (PGD) algorithm .[3]

---

[∗]Equal Contribution
[†]Corresponding Authors

[3]Our codes are available at https://github.com/a1600012888/YOPO-You-Only-Propagate-Once

# 1 Introduction

Deep neural networks achieve state-of-the-art performance on many tasks[4,8,16,21,25,44]. However, recent works show that deep networks are often sensitive to adversarial perturbations[27,35,49], i.e., changing the input in a way imperceptible to humans while causing the neural network to output an incorrect prediction. This poses significant concerns when applying deep neural networks to safety-critical problems such as autonomous driving and medical domains. To effectively defend the adversarial attacks,[26] proposed adversarial training, which can be formulated as a robust optimization[38]:

$$\min_{\theta} \mathbb{E}_{(x,y)\sim\mathcal{D}} \max_{\|\eta\|\leq\epsilon} \ell(\theta; x+\eta, y), \qquad (1)$$

where $\theta$ is the network parameter, $\eta$ is the adversarial perturbation, and $(x,y)$ is a pair of data and label drawn from a certain distribution $\mathcal{D}$. The magnitude of the adversarial perturbation $\eta$ is restricted by $\epsilon > 0$. For a given pair $(x,y)$, we refer to the value of the inner maximization of (1), i.e. $\max_{\|\eta\|\leq\epsilon} \ell(\theta; x+\eta, y)$, as the adversarial loss which depends on $(x,y)$.

A major issue of the current adversarial training methods is their significantly high computational cost. In adversarial training, we need to solve the inner loop, which is to obtain the "optimal" adversarial attack to the input in every iteration. Such "optimal" adversary is usually obtained using multi-step gradient decent, and thus the total time for learning a model using standard adversarial training method is much more than that using the standard training. Considering applying 40 inner iterations of projected gradient descent (PGD[15]) to obtain the adversarial examples, the computation cost of solving the problem (1) is about 40 times that of a regular training.

The main objective of this paper is to reduce the computational burden of adversarial training by limiting the number of forward and backward propagation without hurting the performance of the trained network. In this paper, we exploit the structures that the min-max objectives is encountered with deep neural networks. To achieve this, we formulate the adversarial training problem(1) as a differential game. Afterwards we can derive the Pontryagin's Maximum Principle (PMP) of the problem.

From the PMP, we discover a key fact that the adversarial perturbation is only coupled with the weights of the first layer. This motivates us to propose a novel adversarial training strategy by decoupling the adversary update from the training of the network parameters. This effectively reduces the total number of full forward and backward propagation to only one for each group of adversary updates, significantly lowering the overall computation cost without hampering performance of the trained network.

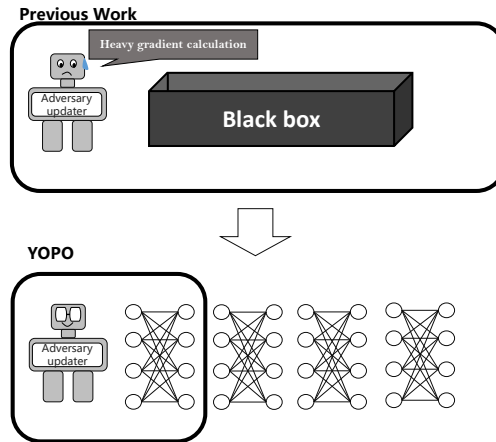

Figure 1: Our proposed YOPO expolits the structure of neural network. To alleviate the heavy computation cost, YOPO focuses the calculation of the adversary at the first layer.

We name this new adversarial training algorithm as **YOPO** (**Y**ou **O**nly **P**ropagate **O**nce). Our numerical experiments show that YOPO achieves approximately 4 ∼5 times speedup over the original PGD adversarial training with comparable accuracy on MNIST/CIFAR10. Furthermore, we apply our algorithm to a recent proposed min max optimization objective "TRADES"[46] and achieve better clean and robust accuracy within half of the time TRADES need.

## 1.1 Related Works

**Adversarial Defense.**   To improve the robustness of neural networks to adversarial examples, many defense strategies and models have been proposed, such as adversarial training[26], orthogonal regularization[6,22], Bayesian method[45], TRADES[46], rejecting adversarial examples[43], Jacobian

regularization[14,29], generative model based defense[12,33], pixel defense[24,31], ordinary differential equation (ODE) viewpoint[47], ensemble via an intriguing stochastic differential equation perspective[39], and feature denoising[34,42], etc. Among all these approaches, adversarial training and its variants tend to be most effective since it largely avoids the the obfuscated gradient problem[2]. Therefore, in this paper, we choose adversarial training to achieve model robustness.

**Neural ODEs.** Recent works have built up the relationship between ordinary differential equations and neural networks[5,10,23,32,37,40,48]. Observing that each residual block of ResNet can be written as $u_{n+1} = u_n + \Delta t f(u_n)$, one step of forward Euler method approximating the ODE $u_t = f(u)$. Thus[19,41] proposed an optimal control framework for deep learning and[5,19,20] utilize the adjoint equation and the maximal principle to train a neural network.

**Decouple Training.** Training neural networks requires forward and backward propagation in a sequential manner. Different ways have been proposed to decouple the sequential process by parallelization. This includes ADMM[36], synthetic gradients[13], delayed gradient[11], lifted machines[1,9,18]. Our work can also be understood as a decoupling method based on a splitting technique. However, we do not attempt to decouple the gradient w.r.t. network parameters but the adversary update instead.

## 1.2 Contribution

- To the best of our knowledge, it is the first attempt to design *NN–specific* algorithm for adversarial defense. To achieve this, we recast the adversarial training problem as a discrete time differential game. From optimal control theory, we derive the an optimality condition, *i.e.* the Pontryagin's Maximum Principle, for the differential game.

- Through PMP, we observe that the adversarial perturbation is only coupled with the first layer of neural networks. The PMP motivates a new adversarial training algorithm, YOPO. We split the adversary computation and weight updating and the adversary computation is focused on the first layer. Relations between YOPO and original PGD are discussed.

- We finally achieve about **4~ 5 times speed up** than the original PGD training with comparable results on MNIST/CIFAR10. Combining YOPO with TRADES[46], we achieve both higher clean and robust accuracy within less than half of the time TRADES need.

## 1.3 Organization

This paper is organized as follows. In Section 2, we formulate the robust optimization for neural network adversarial training as a differential game and propose the gradient based YOPO. In Section 3, we derive the PMP of the differential game, study the relationship between the PMP and the backpropagation based gradient descent methods, and propose a general version of YOPO. Finally, all the experimental details and results are given in Section 4.

## 2 Differential Game Formulation and Gradient Based YOPO

### 2.1 The Optimal Control Perspective and Differential Game

Inspired by the link between deep learning and optimal control[20], we formulate the robust optimization (1) as a differential game[7]. A two-player, zero-sum differential game is a game where each player controls a dynamics, and one tries to maximize, the other to minimize, a payoff functional. In the context of adversarial training, one player is the neural network, which controls the weights of the network to fit the label, while the other is the adversary that is dedicated to producing a false prediction by modifying the input.

The robust optimization problem (1) can be written as a differential game as follows,

$$\min_{\theta} \max_{\|\eta_i\|_{\infty} \leq \epsilon} J(\theta, \eta) := \frac{1}{N} \sum_{i=1}^{N} \ell_i(x_{i,T}) + \frac{1}{N} \sum_{i=1}^{N} \sum_{t=0}^{T-1} R_t(x_{i,t}; \theta_t)$$

$$\text{subject to } x_{i,1} = f_0(x_{i,0} + \eta_i, \theta_0), i = 1, 2, \cdots, N$$

$$x_{i,t+1} = f_t(x_{i,t}, \theta_t), t = 1, 2, \cdots, T-1 \tag{2}$$

Here, the dynamics $\{f_t(x_t, \theta_t), t = 0, 1, \ldots, T-1\}$ represent a deep neural network, $T$ denote the number of layers, $\theta_t \in \Theta_t$ denotes the parameters in layer $t$ (denote $\theta = \{\theta_t\}_t \in \Theta$), the function $f_t : \mathbb{R}^{d_t} \times \Theta_t \to \mathbb{R}^{d_{t+1}}$ is a nonlinear transformation for one layer of neural network where $d_t$ is the dimension of the $t$ th feature map and $\{x_{i,0}, i = 1, \ldots, N\}$ is the training dataset. The variable $\eta = (\eta_1, \cdots, \eta_N)$ is the adversarial perturbation and we constrain it in an $\infty$-ball. Function $\ell_i$ is a data fitting loss function and $R_t$ is the regularization weights $\theta_t$ such as the $L_2$-norm. By casting the problem of adversarial training as a differential game (2), we regard $\theta$ and $\eta$ as two competing players, each trying to minimize/maximize the loss function $J(\theta, \eta)$ respectively.

## 2.2   Gradient Based YOPO

The Pontryagin's Maximum Principle (PMP) is a fundamental tool in optimal control that characterizes optimal solutions of the corresponding control problem[7]. PMP is a rather general framework that inspires a variety of optimization algorithms. In this paper, we will derive the PMP of the differential game (2), which motivates the proposed YOPO in its most general form. However, to better illustrate the essential idea of YOPO and to better address its relations with existing methods such as PGD, we present a special case of YOPO in this section based on gradient descent/ascent. We postpone the introduction of PMP and the general version of YOPO to Section 3.

Let us first rewrite the original robust optimization problem (1) (in a mini-batch form) as

$$\min_\theta \max_{\|\eta_i\| \le \epsilon} \sum_{i=1}^B \ell(g_{\tilde{\theta}}(f_0(x_i + \eta_i, \theta_0)), y_i),$$

where $f_0$ denotes the first layer, $g_{\tilde{\theta}} = f_{T-1}^{\theta_{T-1}} \circ f_{T-2}^{\theta_{T-2}} \circ \cdots f_1^{\theta_1}$ denotes the network without the first layer and $B$ is the batch size. Here $\tilde{\theta}$ is defined as $\{\theta_1, \cdots, \theta_{T-1}\}$. For simplicity we omit the regularization term $R_t$.

The simplest way to solve the problem is to perform gradient ascent on the input data and gradient descent on the weights of the neural network as shown below. Such alternating optimization algorithm is essentially the popular PGD adversarial training[26]. We summarize the PGD-$r$ (for each update on $\theta$) as follows, i.e. performing $r$ iterations of gradient ascent for inner maximization.

- For $s = 0, 1, \ldots, r-1$, perform
  $$\eta_i^{s+1} = \eta_i^s + \alpha_1 \nabla_{\eta_i} \ell(g_{\tilde{\theta}}(f_0(x_i + \eta_i^s, \theta_0)), y_i),\ i = 1, \cdots, B,$$
  where by the chain rule,
  $$\nabla_{\eta_i} \ell(g_{\tilde{\theta}}(f_0(x_i + \eta_i^s, \theta_0)), y_i) = \nabla_{g_{\tilde{\theta}}} \left( \ell(g_{\tilde{\theta}}(f_0(x_i + \eta_i^s, \theta_0)), y_i) \right) \cdot$$
  $$\nabla_{f_0} \left( g_{\tilde{\theta}}(f_0(x_i + \eta_i^s, \theta_0)) \right) \cdot \nabla_{\eta_i} f_0(x_i + \eta_i^s, \theta_0).$$
- Perform the SGD weight update (momentum SGD can also be used here)
  $$\theta \leftarrow \theta - \alpha_2 \nabla_\theta \left( \sum_{i=1}^B \ell(g_{\tilde{\theta}}(f_0(x_i + \eta_i^m, \theta_0)), y_i) \right)$$

Note that this method conducts $r$ sweeps of forward and backward propagation for each update of $\theta$. This is the main reason why adversarial training using PGD-type algorithms can be very slow.

To reduce the total number of forward and backward propagation, we introduce a slack variable

$$p = \nabla_{g_{\tilde{\theta}}} \left( \ell(g_{\tilde{\theta}}(f_0(x_i + \eta_i, \theta_0)), y_i) \right) \cdot \nabla_{f_0} \left( g_{\tilde{\theta}}(f_0(x_i + \eta_i, \theta_0)) \right)$$

and freeze it as a constant within the inner loop of the adversary update. The modified algorithm is given below and we shall refer to it as YOPO-$m$-$n$.

- Initialize $\{\eta_i^{1,0}\}$ for each input $x_i$. For $j = 1, 2, \cdots, m$
  - Calculate the slack variable $p$
  $$p = \nabla_{g_{\tilde{\theta}}}\left(\ell(g_{\tilde{\theta}}(f_0(x_i + \eta_i^{j,0}, \theta_0)), y_i)\right) \cdot \nabla_{f_0}\left(g_{\tilde{\theta}}(f_0(x_i + \eta_i^{j,0}, \theta_0))\right),$$
  - Update the adversary for $s = 0, 1, \ldots, n-1$ for fixed $p$
  $$\eta_i^{j,s+1} = \eta_i^{j,s} + \alpha_1 p \cdot \nabla_{\eta_i} f_0(x_i + \eta_i^{j,s}, \theta_0), i = 1, \cdots, B$$
  - Let $\eta_i^{j+1,0} = \eta_i^{j,n}$.
- Calculate the weight update
$$U = \sum_{j=1}^{m} \nabla_\theta \left(\sum_{i=1}^{B} \ell(g_{\tilde{\theta}}(f_0(x_i + \eta_i^{j,n}, \theta_0)), y_i)\right)$$
and update the weight $\theta \leftarrow \theta - \alpha_2 U$. (Momentum SGD can also be used here.)

Intuitively, YOPO freezes the values of the derivatives of the network at level $1, 2 \ldots, T-1$ during the $s$-loop of the adversary updates. Figure 2 shows the conceptual comprison between YOPO and PGD. YOPO-$m$-$n$ accesses the data $m \times n$ times while only requires $m$ full forward and backward propagation. PGD-$r$, on the other hand, propagates the data $r$ times for $r$ full forward and backward propagation. As one can see that, YOPO-$m$-$n$ has the flexibility of increasing $n$ and reducing $m$ to achieve approximately the same level of attack but with much less computation cost. For example, suppose one applies PGD-10 (i.e. 10 steps of gradient ascent for solving the inner maximization) to calculate the adversary. An alternative approach is using YOPO-5-2 which also accesses the data 10 times but the total number of full forward propagation is only 5. Empirically, YOPO-m-n achieves comparable results only requiring setting $m \times n$ a litter larger than $r$.

Another benefit of YOPO is that we take full advantage of every forward and backward propagation to update the weights, i.e. the intermediate perturbation $\eta_i^j, j = 1, \cdots, m-1$ are not wasted like PGD-$r$. This allows us to perform multiple updates per iteration, which potentially drives YOPO to converge faster in terms of the number of epochs. Combining the two factors together, YOPO significantly could accelerate the standard PGD adversarial training.

We would like to point out a concurrent paper[30] that is related to YOPO. Their proposed method, called "Free-$m$", also can significantly speed up adversarial training. In fact, Free-$m$ is essentially YOPO-$m$-1, except that YOPO-$m$-1 delays the weight update after the whole mini-batch is processed in order for a proper usage of momentum [4].

## 3 The Pontryagin's Maximum Principle for Adversarial Training

In this section, we present the PMP of the discrete time differential game (2). From the PMP, we can observe that the adversary update and its associated back-propagation process can be decoupled. Furthermore, back-propagation based gradient descent can be understood as an iterative algorithm solving the PMP and with that the version of YOPO presented in the previous section can be viewed as an algorithm solving the PMP. However, the PMP facilitates a much wider class of algorithms than gradient descent algorithms[19]. Therefore, we will present a general version of YOPO based on the PMP for the discrete differential game.

### 3.1 PMP

Pontryagin type of maximal principle[3,28] provides necessary conditions for optimality with a layer-wise maximization requirement on the Hamiltonian function. For each layer $t \in [T] :=$

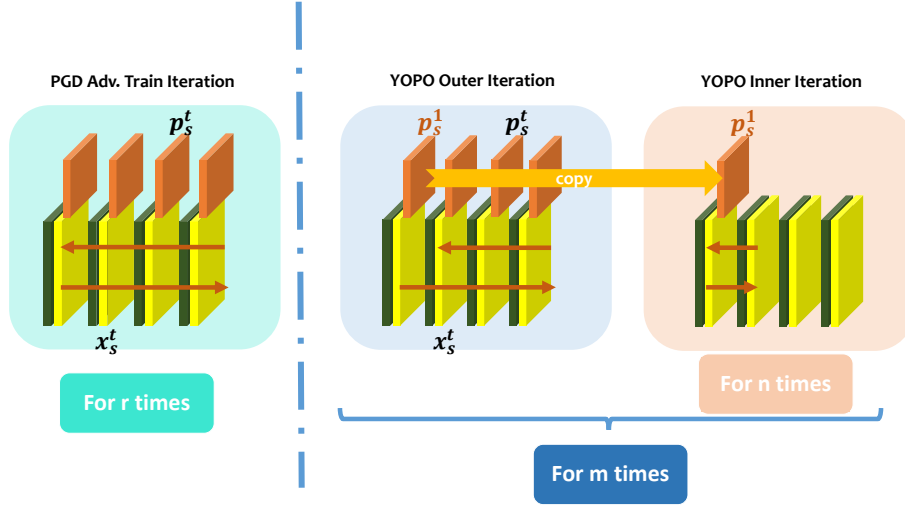

Figure 2: Pipeline of YOPO-$m$-$n$ described in Algorithm 1. The yellow and olive blocks represent feature maps while the orange blocks represent the gradients of the loss w.r.t. feature maps of each layer.

$\{0, 1, \ldots, T-1\}$, we define the Hamiltonian function $H_t : \mathbb{R}^{d_t} \times \mathbb{R}^{d_{t+1}} \times \Theta_t \to \mathbb{R}$ as

$$H_t(x, p, \theta_t) = p \cdot f_t(x, \theta_t) - \frac{1}{B} R_t(x, \theta_t).$$

The PMP for continuous time differential game has been well studied in the literature [7]. Here, we present the PMP for our discrete time differential game (2).

**Theorem 1.** *(PMP for adversarial training) Assume $\ell_i$ is twice continuous differentiable, $f_t(\cdot, \theta), R_t(\cdot, \theta)$ are twice continuously differentiable with respect to $x$; $f_t(\cdot, \theta), R_t(\cdot, \theta)$ together with their $x$ partial derivatives are uniformly bounded in $t$ and $\theta$, and the sets $\{f_t(x, \theta) : \theta \in \Theta_t\}$ and $\{R_t(x, \theta) : \theta \in \Theta_t\}$ are convex for every $t$ and $x \in \mathbb{R}^{d_t}$. Denote $\theta^*$ as the solution of the problem (2), then there exists co-state processes $p_i^* := \{p_{i,t}^* : t \in [T]\}$ such that the following holds for all $t \in [T]$ and $i \in [B]$:*

$$x_{i,t+1}^* = \nabla_p H_t(x_{i,t}^*, p_{i,t+1}^*, \theta_t^*), \qquad\qquad x_{i,0}^* = x_{i,0} + \eta_i^* \qquad (3)$$

$$p_{i,t}^* = \nabla_x H_t(x_{i,t}^*, p_{i,t+1}^*, \theta_t^*), \qquad\qquad p_{i,T}^* = -\frac{1}{B} \nabla \ell_i(x_{i,T}^*) \qquad (4)$$

*At the same time, the parameters of the first layer $\theta_0^* \in \Theta_0$ and the optimal adversarial perturbation $\eta_i^*$ satisfy*

$$\sum_{i=1}^{B} H_0(x_{i,0}^* + \eta_i, p_{i,1}^*, \theta_0^*) \geq \sum_{i=1}^{B} H_0(x_{i,0}^* + \eta_i^*, p_{i,1}^*, \theta_0^*) \geq \sum_{i=1}^{B} H_0(x_{i,0}^* + \eta_i^*, p_{i,1}^*, \theta_0), \quad (5)$$

$$\forall \theta_0 \in \Theta_0, \|\eta_i\|_\infty \leq \epsilon \quad (6)$$

*and the parameters of the other layers $\theta_t^* \in \Theta_t, t \in [T]$ maximize the Hamiltonian functions*

$$\sum_{i=1}^{B} H_t(x_{i,t}^*, p_{i,t+1}^*, \theta_t^*) \geq \sum_{i=1}^{B} H_t(x_{i,t}^*, p_{i,t+1}^*, \theta_t), \ \forall \theta_t \in \Theta_t \qquad (7)$$

*Proof.* Proof is in the supplementary materials. $\qquad\qquad\qquad\qquad\qquad\qquad\qquad\qquad\square$

From the theorem, we can observe that the adversary $\eta$ is only coupled with the parameters of the first layer $\theta_0$. This key observation inspires the design of YOPO.

## 3.2 PMP and Back-Propagation Based Gradient Descent

The classical back-propagation based gradient descent algorithm[17] can be viewed as an algorithm attempting to solve the PMP. Without loss of generality, we can let the regularization term $R = 0$, since we can simply add an extra dynamic $w_t$ to evaluate the regularization term $R$, *i.e.*

$$w_{t+1} = w_t + R_t(x_t, \theta_t), \; w_0 = 0.$$

We append $w$ to $x$ to study the dynamics of a new $(d_t + 1)$-dimension vector and change $f_t(x, \theta_t)$ to $(f_t(x, \theta_t), w + R_t(x, \theta_t))$. The relationship between the PMP and the back-propagation based gradient descent method was first observed by Li et al. [19]. They showed that the forward dynamical system Eq.(3) is the same as the neural network forward propagation. The backward dynamical system Eq.(4) is the back-propagation, which is formally described by the following lemma.

**Lemma 1.**

$$p_t^* = \nabla_x H_t(x_t^*, p_{t+1}^*, \theta_t^*) = \nabla_x f(x_t^*, \theta_t^*)^T p_{t+1} = (\nabla_{x_t} x_{t+1}^*)^T \cdot -\nabla_{x_{t+1}}(\ell(x_T)) = -\nabla_{x_t}(\ell(x_T)).$$

To solve the maximization of the Hamiltonian, a simple way is the gradient ascent:

$$\theta_t^1 = \theta_t^0 + \alpha \cdot \nabla_\theta \sum_{i=1}^{B} H_t(x_{i,t}^{\theta^0}, p_{i,t+1}^{\theta^0}, \theta_t^0). \tag{8}$$

**Theorem 2.** *The update (8) is equivalent to gradient descent method for training networks[19,20].*

## 3.3 YOPO from PMP's View Point

Based on the relationship between back-propagation and the Pontryagin's Maximum Principle, in this section, we provide a new understanding of YOPO, *i.e.* solving the PMP for the differential game. Observing that, in the PMP, the adversary $\eta$ is only coupled with the weight of the first layer $\theta_0$. Thus we can update the adversary via minimizing the Hamiltonian function instead of directly attacking the loss function, described in Algorithm 1.

For YOPO-$m$-$n$, to approximate the exact minimization of the Hamiltonian, we perform $n$ times gradient descent to update the adversary. Furthermore, in order to make the calculation of the adversary more accurate, we iteratively pass one data point $m$ times. Besides, the network weights are optimized via performing the gradient ascent to Hamiltonian, resulting in the gradient-based YOPO proposed in Section 2.2.

# 4 Experiments

## 4.1 YOPO for Adversarial Training

To demonstrate the effectiveness of YOPO, we conduct experiments on MNIST and CIFAR10. We find that models trained with YOPO have comparable performance with that of the PGD adversarial training, but with a much fewer computational cost. We also compare our method with a concurrent method "For Free"[30], and the result shows that our algorithm can achieve comparable performance with around 2/3 GPU time of their official implementation.

**MNIST**  We achieve comparable results with the best in [5] within 250 seconds, while it takes PGD-40 more than 1250s to reach the same level. The accuracy-time curve is shown in Figure 3(a). Quantitative results can be seen in supplementary materials. Naively reducing the backprop times of PGD-40 to PGD-10 will harm the robustness, as can be seen in supplementary materials.

**CIFAR10.**  [26] performs a 7-step PGD to generate adversary while training. As a comparison, we test YOPO-3-5 and YOPO-5-3 with a step size of 2/255. Quantitative results can be seen in Table 1 and supplementary materials.

Under PreAct-Res18, for YOPO-5-3, it achieves comparable robust accuracy with[26] with around half computation for every epoch. The accuracy-time curve is shown in Figure 3(b).The quantitative results can be seen in supplementary materials.

**Algorithm 1** YOPO (**Y**ou **O**nly **P**ropagate **O**nce)

---

Randomly initialize the network parameters or using a pre-trained network.
**repeat**
    Randomly select a mini-batch $\mathcal{B} = \{(x_1, y_1), \cdots, (x_B, y_B)\}$ from training set.
    Initialize $\eta_i, i = 1, 2, \cdots, B$ by sampling from a uniform distribution between [-$\epsilon$, $\epsilon$]
    **for** $j = 1$ to $m$ **do**
        $x_{i,0} = x_i + \eta_i^j, i = 1, 2, \cdots, B$
        **for** $t = 0$ to $T - 1$ **do**
            $x_{i,t+1} = \nabla_p H_t(x_{i,t}, p_{i,t+1}, \theta_t), i = 1, 2, \cdots, B$
        **end for**
        $p_{i,T} = -\frac{1}{B} \nabla \ell(x_{i,T}^*), i = 1, 2, \cdots, B$
        **for** $t = T - 1$ to $0$ **do**
            $p_{i,t} = \nabla_x H_t(x_{i,t}, p_{i,t+1}, \theta_t), i = 1, 2, \cdots, B$
        **end for**
        $\eta_i^j = \arg\min_{\eta_i} H_0(x_{i,0} + \eta_i, p_{i,0}, \theta_0), i = 1, 2, \cdots, B$
    **end for**
    **for** $t = T - 1$ to $1$ **do**
        $\theta_t = \arg\max_{\theta_t} \sum_{i=1}^{B} H_t(x_{i,t}, p_{i,t+1}, \theta_t)$
    **end for**
    $\theta_0 = \arg\max_{\theta_0} \frac{1}{m} \sum_{k=1}^{m} \sum_{i=1}^{B} H_0(x_{i,0} + \eta_i^j, p_{i,1}, \theta_0)$
**until** Convergence

---

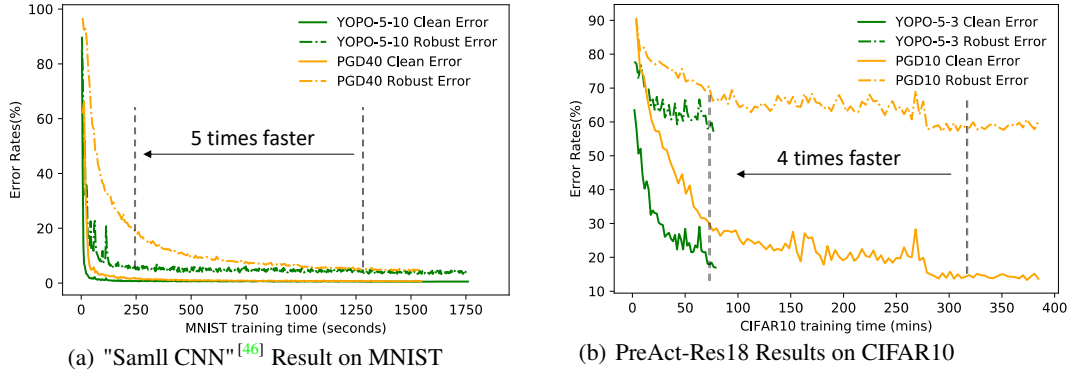

    (a) "Samll CNN"[46] Result on MNIST          (b) PreAct-Res18 Results on CIFAR10

Figure 3: Performance w.r.t. training time

As for Wide ResNet34, YOPO-5-3 still achieves similar acceleration against PGD-10, as shown in Table 1. We also test PGD-3/5 to show that naively reducing backward times for this minmax problem[26] cannot produce comparable results within the same computation time as YOPO. Meanwhile, YOPO-3-5 can achieve more aggressive speed-up with only a slight drop in robustness.

## 4.2 YOPO for TRADES

TRADES[46] formulated a new min-max objective function of adversarial defense and achieves the state-of-the-art adversarial defense results. The experiment details are in supplementary material, and quantitative results are demonstrated in Table 2.

| Training Methods | Clean Data | PGD-20 Attack | CW Attack | Training Time (mins) |
|---|---|---|---|---|
| TRADES-10[46] | 86.14% | 44.50% | 58.40% | 633 |
| TRADES-YOPO-3-4 (Ours) | 87.82% | 46.13% | 59.48% | 259 |
| TRADES-YOPO-2-5 (Ours) | 88.15% | 42.48% | 59.25% | 218 |

Table 2: Results of training PreAct-Res18 for CIFAR10 with TRADES objective

| Training Methods | Clean Data | PGD-20 Attack | Training Time (mins) |
|---|---|---|---|
| Natural train | 95.03% | 0.00% | 233 |
| PGD-3[26] | 90.07% | 39.18% | 1134 |
| PGD-5[26] | 89.65% | 43.85% | 1574 |
| PGD-10[26] | 87.30% | 47.04% | 2713 |
| Free-8[30]1 | 86.29% | 47.00% | 667 |
| YOPO-3-5 (Ours) | 87.27% | 43.04% | 299 |
| YOPO-5-3 (Ours) | 86.70% | 47.98% | **476** |

1 Code from https://github.com/ashafahi/free_adv_train.
Table 1: Results of Wide ResNet34 for CIFAR10.

## 5 Conclusion

In this work, we have developed an efficient strategy for accelerating adversarial training. We recast the adversarial training of deep neural networks as a discrete time differential game and derive a Pontryagin's Maximum Principle (PMP) for it. Based on this maximum principle, we discover that the adversary is only coupled with the weights of the first layer. This motivates us to split the adversary updates from the back-propagation gradient calculation. The proposed algorithm, called YOPO, avoids computing full forward and backward propagation for too many times, thus effectively reducing the computational time as supported by our experiments.

## Acknowledgement

We thank Di He and Long Chen for beneficial discussion. Zhanxing Zhu is supported in part by National Natural Science Foundation of China (No.61806009), Beijing Natural Science Foundation (No. 4184090) and Beijing Academy of Artificial Intelligence (BAAI). Bin Dong is supported in part by Beijing Natural Science Foundation (No. Z180001) and Beijing Academy of Artificial Intelligence (BAAI). Dinghuai Zhang is supported by the Elite Undergraduate Training Program of Applied Math of the School of Mathematical Sciences at Peking University.

## Footnotes

[4]Momentum should be accumulated between mini-batches other than different adversarial examples from one mini-batch, otherwise overfitting will become a serious problem.

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
