[Supplementary Material · YOPO_NeurIPS2019_supplementary.pdf]

# Supplementary Materials:
# You Only Propagate Once: Accelerating Adversarial Training via Maximal Principle

## A    Proof Of The Theorems

### A.1    Proof of Theorem 1

In this section we give the full statement of the maximum principle for the adversarial training and present a proof. Let's start from the case of the natural training of neural networks.

**Theorem.** *(PMP for adversarial training) Assume $\ell_i$ is twice continuous differentiable, $f_t(\cdot, \theta), R_t(\cdot, \theta)$ are twice continuously differentiable with respect to $x$, and $f_t(\cdot, \theta), R_t(\cdot, \theta)$ together with their $x$ partial derivatives are uniformly bounded in $t$ and $\theta$. The sets $\{f_t(x, \theta) : \theta \in \Theta_t\}$ and $\{R_t(x, \theta) : \theta \in \Theta_t\}$ are convex for every $t$ and $x \in \mathbb{R}^{d_t}$. Let $\theta^*$ to be the solution of*

$$\min_{\theta \in \Theta} \max_{\|\eta\|_\infty \leq \epsilon} J(\theta, \eta) := \frac{1}{N} \sum_{i=1}^N \ell_i(x_{i,T}) + \frac{1}{N} \sum_{i=1}^N \sum_{t=0}^{T-1} R_t(x_{i,t}, \theta_t) \tag{1}$$

$$\textit{subject to } x_{i,1} = f_0(x_{i,0} + \eta_i; \theta_0), i = 1, 2, \cdots, N \tag{2}$$

$$x_{i,t+1} = f_t(x_{i,t}, \theta_t), t = 1, 2, \cdots, T - 1. \tag{3}$$

*Then there exists co-state processes $p_i^* := p_{i,t}^* : t = 0, \cdots, T$ such that the following holds for all $t \in [T]$ and $i \in [N]$:*

$$x_{i,t+1}^* = \nabla_p H_t(x_{i,t}^*, p_{i,t+1}^*, \theta_t^*), \qquad\qquad x_{i,0}^* = x_{i,0} + \eta_i^* \tag{4}$$

$$p_{i,t}^* = \nabla_x H_t(x_{i,t}^*, p_{i,t+1}^*, \theta_t^*), \qquad\qquad p_{i,T}^* = -\frac{1}{N} \nabla \ell_i(x_{i,T}^*) \tag{5}$$

*Here $H$ is the per-layer defined Hamiltonian function $H_t : \mathbb{R}^{d_t} \times \mathbb{R}^{d_{t+1}} \times \Theta_t \to \mathbb{R}$ as*

$$H_t(x, p, \theta_t) = p \cdot f_t(x, \theta_t) - \frac{1}{N} R_t(x, \theta_t)$$

*At the same time, the parameter of the first layer $\theta_0^* \in \Theta_0$ and the best perturbation $\eta^*$ satisfy*

$$\sum_{i=1}^N H_0(x_{i,0}^* + \eta_i, p_{i,1}^*, \theta_0^*) \geq \sum_{i=1}^N H_0(x_{i,0}^* + \eta_i^*, p_{i,1}^*, \theta_0^*) \geq \sum_{i=1}^N H_0(x_{i,0}^* + \eta_i^*, p_{i,1}^*, \theta_0), \forall \theta_0 \in \Theta_0, \|\eta_i\|_\infty \leq \epsilon \tag{6}$$

*while parameter of the other layers $\theta_t^* \in \Theta_t, t = 1, 2, \cdots, T - 1$ will maximize the Hamiltonian functions*

$$\sum_{i=1}^N H_t(x_{i,t}^*, p_{i,t+1}^*, \theta_t^*) \geq \sum_{i=1}^N H_t(x_{i,t}^*, p_{i,t+1}^*, \theta_t), \forall \theta_t \in \Theta_t \tag{7}$$

*Proof.* We first propose PMP for discrete time dynamic system and utilize it directly gives out the proof of PMP for adversarial training.

**Lemma 1.** *(PMP for discrete time dynamic system) Assume $\ell$ is twice continuous differentiable, $f_t(\cdot, \theta), R_t(\cdot, \theta)$ are twice continuously differentiable with respect to $x$, and $f_t(\cdot, \theta), R_t(\cdot, \theta)$ together with their $x$ partial derivatives are uniformly bounded in $t$ and $\theta$. The sets $\{f_t(x, \theta) : \theta \in \Theta_t\}$ and $\{R_t(x, \theta) : \theta \in \Theta_t\}$ are convex for every $t$ and $x \in \mathbb{R}^{d_t}$. Let $\theta^*$ to be the solution of*

$$\min_{\theta \in \Theta} \max_{\|\eta\|_\infty \leq \epsilon} J(\theta, \eta) := \frac{1}{N} \sum_{i=1}^N \ell_i(x_{i,T}) + \frac{1}{N} \sum_{i=1}^N \sum_{t=0}^{T-1} R_t(x_{i,t}, \theta_t) \tag{8}$$

$$\text{subject to } x_{i,t+1} = f_t(x_{i,t}, \theta_t), i \in [N], t = 0, 1, \cdots, T-1. \tag{9}$$

*There exists co-state processes $p_i^* := p_{i,t}^* : t = 0, \cdots, T$ such that the following holds for all $t \in [T]$ and $i \in [N]$:*

$$x_{i,t+1}^* = \nabla_p H_t(x_{i,t}^*, p_{i,t+1}^*, \theta_t^*), \qquad\qquad x_{i,0}^* = x_{i,0} \tag{10}$$

$$p_{i,t}^* = \nabla_x H_t(x_{i,t}^*, p_{i,t+1}^*, \theta_t^*), \qquad\qquad p_{i,T}^* = -\frac{1}{N}\nabla \ell_i(x_{i,T}^*) \tag{11}$$

*Here $H$ is the per-layer defined Hamiltonian function $H_t : \mathbb{R}^{d_t} \times \mathbb{R}^{d_{t+1}} \times \Theta_t \to \mathbb{R}$ as*

$$H_t(x, p, \theta_t) = p \cdot f_t(x, \theta_t) - \frac{1}{N} R_t(x, \theta_t)$$

*The parameters of the layers $\theta_t^* \in \Theta_t, t = 0, 1, \cdots, T-1$ will maximize the Hamiltonian functions*

$$\sum_{i=1}^N H_t(x_{i,t}^*, p_{i,t+1}^*, \theta_t^*) \geq \sum_{i=1}^N H_t(x_{i,t}^*, p_{i,t+1}^*, \theta_t), \forall \theta_t \in \Theta_t \tag{12}$$

*Proof.* Without loss of generality, we let $L = 0$. The reason is that we can simply add an extra dynamic $w_t$ to calculate the regularization term $R$, *i.e.*

$$w_{t+1} = w_t + R_t(x_t, \theta_t), w_0 = 0.$$

We append $w$ to $x$ to study the dynamic of a new $d_t + 1$ dimension vector and modify $f_t(x, \theta)$ to $(f_t(x, \theta), w + R_t(x, \theta))$. Thus we only need to prove the case when $L = 0$.

For simplicity, we omit the subscript $s$ in the following proof. (Concatenating all $x_s$ into $x = (x_1, \ldots, x_N)$ can justify this.)

Now we begin the proof. Following the linearization lemma in [**?** ] [19], consider the linearized problem

$$\phi_{t+1} = f_t(x_t^*, \theta_t) + \nabla_x f_t(x_t^*, \theta_t)(\phi_t - x_t^*), \phi_0 = x_0 + \eta. \tag{13}$$

The reachable states by the linearized dynamic system is denoted as

$$W_t := \{x \in \mathbb{R}^{d_t} : \exists \theta, \eta = \eta^* \text{ s.t. } \phi_t^\theta = x\}$$

here $x_t^\theta$ denotes the the evolution of the dynamical system for $x_t$ under $\theta$. We also define

$$S := \{x \in \mathbb{R}^{d_T} : (x - x_T^*)\nabla \ell(x_T^*) < 0\}$$

The linearization lemma in [**?** 19] tells us that $W_T$ and $S$ are separated by $\{x : p_T^* \cdot (x - x_T^*) = 0, p_T^* = -\nabla \ell(x_T^*)\}$, *i.e.*

$$p_T^* \cdot (x - x_T^*) \leq 0, \forall x \in W_t. \tag{14}$$

Thus setting

$$p_t^* = \nabla_x H_t(x_t^*, p_{t+1}^*, \theta_t^*) = \nabla_x f(x_t^*, \theta_t^*)^T \cdot p_{t+1}^*,$$

we have

$$(\phi_{t+1} - x_{t+1}^*) \cdot p_t^* = (\phi_t - x_t^*) \cdot p_t^*. \tag{15}$$

Thus from Eq.14 and Eq.15 we get

$$p_{t+1}^* \cdot (\phi_{t+1}^\theta - x_{t+1}^*) \leq 0, \quad t = 0, \cdots, T-1, \forall \theta \in \Theta := \Theta_0 \times \Theta_1 \times \cdots$$

Setting $\theta_s = \theta_s^*$ for $s < t$ we have $\phi_{t+1}^\theta = f_t(x_t^*, \theta_t)$, which leads to $p_{t+1}^* \cdot (f_t(x_t^*, \theta_t) - x_{t+1}^*) \leq 0$. This finishes the proof of the maximal principle on weight space $\Theta$.

$\square$

We return to the proof of the theorem. The proof of the maximal principle on the weight space, *i.e.*

$$\sum_{i=1}^{N} H_t(x_{i,t}^*, p_{i,t+1}^*, \theta_t^*) \geq \sum_{i=1}^{N} H_t(x_{i,t}^*, p_{i,t+1}^*, \theta), \forall \theta_t \in \Theta_t, t = 1, 2, \cdots, T-1$$

and

$$\sum_{i=1}^{N} H_0(x_{i,0}^* + \eta_i^*, p_{i,1}^*, \theta_0^*) \geq \sum_{i=1}^{N} H_0(x_{i,0}^* + \eta_i^*, p_{i,1}^*, \theta_0), \forall \theta_0 \in \Theta_0,$$

can be reached with the help of Lemma 1: replacing the dynamic start point $x_{i,0}$ in Eq.10 with $x_{i,0} + \eta_i^*$ makes this maximal principle a direct corollary of Lemma 1.

Next, we prove the Hamiltonian conidition for the adversary, *i.e.*

$$\sum_{i=1}^{N} H_0(x_{i,0}^* + \eta_i^*, p_{i,1}^*, \theta_0^*) \leq \sum_{i=1}^{N} H_0(x_{i,0}^* + \eta_i, p_{i,1}^*, \theta_0^*), \forall \|\eta_i\|_\infty \leq \epsilon \qquad (16)$$

Assuming $R_{i,t} = 0$ like above, we define a new optimal control problem with target function $\tilde{\ell}_i() = -\ell_i()$ and previous dynamics except $x_{i,1} = \tilde{f}_0(x_{i,0}; \theta_0, \eta_i) = f_0(x_{i,0} + \eta_i; \theta_0)$:

$$\min_{\|\eta\|_\infty \leq \epsilon} \tilde{J}(\theta, \eta) := \frac{1}{N} \sum_{i=1}^{N} \tilde{\ell}_i(x_{i,T}) \qquad (17)$$

$$\text{subject to } x_{i,1} = \tilde{f}_0(x_{i,0}; \theta_0, \eta_i), i = 1, 2, \cdots, N \qquad (18)$$

$$x_{i,t+1} = f_t(x_{i,t}, \theta_t), t = 1, 2, \cdots, T-1. \qquad (19)$$

However in this time, all the layer parameters $\theta_t$ are **fixed** and $\eta_i$ is the control. From the above Lemma 1 we get

$$\tilde{x}_{i,1}^* = \nabla_p \tilde{H}_0(\tilde{x}_{i,0}^*, \tilde{p}_{i,1}^*, \theta_0, \eta_i^*), \quad \tilde{x}_{i,t+1}^* = \nabla_p H_t(\tilde{x}_{i,t}^*, \tilde{p}_{i,t+1}^*, \theta_t), \quad \tilde{x}_{i,0}^* = x_{i,0}, \qquad (20)$$

$$\tilde{p}_{i,0}^* = \nabla_x \tilde{H}_0(\tilde{x}_{i,0}^*, \tilde{p}_{i,1}^*, \theta_0, \eta_i^*), \qquad \tilde{p}_{i,t}^* = \nabla_x H_t(\tilde{x}_{i,t}^*, \tilde{p}_{i,t+1}^*, \theta_t), \quad \tilde{p}_{i,T}^* = \frac{1}{N} \nabla \ell_i(\tilde{x}_{i,T}^*), \quad (21)$$

where $\tilde{H}_0(x, p, \theta_0, \eta) = p \cdot \tilde{f}_0(x; \theta_0, \eta) = p \cdot f_0(x + \eta; \theta_0)$ and $t = 1, \cdots, T-1$. This gives the fact that $\tilde{x}_{i,t}^* = x_{i,t}^*$. Lemma 1 also tells us

$$\sum_{i=1}^{N} \tilde{H}_0(\tilde{x}_{i,0}^*, \tilde{p}_{i,t+1}^*, \theta_0, \eta_i^*) \geq \sum_{i=1}^{N} \tilde{H}_0(\tilde{x}_{i,0}^*, \tilde{p}_{i,1}^*, \theta_0, \eta_i), \forall \|\eta_i\|_\infty \leq \epsilon \qquad (22)$$

which is

$$\sum_{i=1}^{N} \tilde{p}_{i,1}^* \cdot f_0(\tilde{x}_{i,0}^* + \eta_i^*; \theta_0) \geq \sum_{i=1}^{N} \tilde{p}_{i,1}^* \cdot f_0(\tilde{x}_{i,0}^* + \eta_i; \theta_0), \forall \|\eta_i\|_\infty \leq \epsilon \qquad (23)$$

On the other hand, Lemma 1 gives

$$\tilde{p}_t^* = -\nabla_{x_t}(\tilde{\ell}(x_T)) = \nabla_{x_t}(\ell(x_T)) = -p_t^*.$$

Then we have

$$\sum_{i=1}^{N} p_{i,1}^* \cdot f_0(x_{i,0}^* + \eta_i^*; \theta_0) \leq \sum_{i=1}^{N} p_{i,1}^* \cdot f_0(x_{i,0}^* + \eta_i; \theta_0), \forall \|\eta_i\|_\infty \leq \epsilon \qquad (24)$$

which is

$$\sum_{i=1}^{N} H_0(x_{i,0}^*, p_{i,t+1}^*, \theta_0, \eta_i^*) \leq \sum_{i=1}^{N} H_0(x_{i,0}^*, p_{i,1}^*, \theta_0, \eta_i), \forall \|\eta_i\|_\infty \leq \epsilon \qquad (25)$$

This finishes the proof for the adversarial control.

$\square$

**Remark.** *The additional assumption that the sets $\{f_t(x, \theta) : \theta \in \Theta_t\}$ and $\{R_t(x, \theta) : \theta \in \Theta_t\}$ are convex for every $t$ and $x \in \mathbb{R}^{d_t}$ is extremely weak and is not unrealistic which is already explained in [19].*

# B Experiment Setup and Supplementary Experiments

## B.1 MNIST

Training against PGD-40 is a common practice to get sota results on MNIST. We adopt network architectures from [42] with four convolutional layers followed by three fully connected layers. Following [42] and [23], we set the size of perturbation as $\epsilon = 0.3$ in an infinite norm sense. Experiments are taken on idle NVIDIA Tesla P100 GPUs. We train models for 55 epochs with a batch size of 256, longer than what convergence needs for both training methods. The learning rate is set to 0.1 initially, and is lowered by 10 times at epoch 45. We use a weight decay of $5e - 4$ and a momentum of $0.9$. To measure the robustness of trained models, we performed a PGD-40 and CW[?] attack with CW coefficient $c = 5e2$ and $lr = 1e - 2$.

| Training Methods | Clean Data | PGD-40 Attack | CW Attack |
|:---:|:---:|:---:|:---:|
| PGD-5 [23] | 99.43% | 42.39% | 77.04% |
| PGD-10 [23] | 99.53% | 77.00% | 82.00% |
| PGD-40 [23] | 99.49% | 96.56% | 93.52% |
| YOPO-5-10 (Ours) | 99.46% | 96.27% | 93.56% |

Table 1: Results of MNIST robust training. YOPO-5-10 achieves state-of-the-art result as PGD-40. Notice that for every epoch, PGD-5 and YOPO-5-3 have approximately the same computational cost.

## B.2 CIFAR-10

Following [23], we take Preact-ResNet18 and Wide ResNet-34 as the models for testing. We set the the size of perturbation as $\epsilon = 8/255$ in an infinite norm sense. We perform a 20 steps of PGD with step size $2/255$ when testing. For PGD adversarial training, we train models for 105 epochs as a common practice. The learning rate is set to $5e - 2$ initially, and is lowered by 10 times at epoch 79, 90 and 100. For YOPO-$m$-$n$, we train models for 40 epochs which is much longer than what convergence needs. The learning rate is set to $0.2/m$ initially, and is lowered by 10 times at epoch 30 and 36. We use a batch size of 256, a weight decay of $5e - 4$ and a momentum of $0.9$ for both algorithm. We also test our model's robustness under CW attack [? ] with $c = 5e2$ and $lr = 1e - 2$. The experiments are taken on idle NVIDIA GeForce GTX 1080 Ti GPUs.

| Training Methods | Clean Data | PGD-20 Attack | CW Attack |
|:---:|:---:|:---:|:---:|
| PGD-3 [23] | 88.19% | 32.51% | 54.65% |
| PGD-5 [23] | 86.63% | 37.78% | 57.71% |
| PGD-10 [23] | 84.82% | 41.61% | 58.88% |
| YOPO-3-5 (Ours) | 82.14% | 38.18% | 55.73% |
| YOPO-5-3 (Ours) | 83.99% | 44.72% | 59.77% |

Table 2: Results of PreAct-Res18 for CIFAR10. Note that for every epoch, PGD-3 and YOPO-3-5 have the approximately same computational cost, and so do PGD-5 and YOPO-5-3.

## B.3 TRADES

TRADES[42] achieves the state-of-the-art results in adversarial defensing. The methodology achieves the 1st place out of the 1,995 submissions in the robust model track of NeurIPS 2018 Adversarial Vision Challenge. TRADES proposed a surrogate loss which quantify the trade-off in terms of the gap between the risk for adversarial examples and the risk for non-adversarial examples and the objective function can be formulated as

$$\min_{\theta} \mathbb{E}_{(x,y)\sim\mathcal{D}} \max_{\|\eta\|\leq\epsilon} \left(\ell(f_{\theta}(x), y) + \mathcal{L}\left(f_{\theta}\left(x\right), f_{\theta}\left(x + \eta\right)\right)/\lambda\right) \tag{26}$$

where $f_{\theta}(x)$ is the neural network parameterized by $\theta$, $\ell$ denotes the loss function, $\mathcal{L}(\cdot, \cdot)$ denotes the consistency loss and $\lambda$ is a balancing hyper parameter which we set to be 1 as in [42]. To solve the min-max problem, [42] also searched the ascent direction via the gradient of the "adversarial loss", *i.e.* generating the adversarial example before performing gradient descent on the weight. Specifically,

the PGD attack is performed to maximize a consistency loss instead of classification loss. For each clean data $x$, a single iteration of the adversarial attach can be formulated as

$$x' \leftarrow \Pi_{\|x'-x\| \leq \epsilon} \left( \alpha_1 \operatorname{sign} \left( \nabla_{x'} \mathcal{L} \left( f_\theta \left( x \right), f_\theta \left( x' \right) \right) \right) + x' \right),$$

where $\Pi$ is projection operator. In the implementation of [42], after 10 such update iterations for each input data $x_i$, the update for weights is performed as

$$\theta \leftarrow \theta - \alpha_2 \sum_{i=1}^{B} \nabla_\theta \left[ \ell \left( f_\theta \left( x_i \right), y_i \right) + \mathcal{L} \left( f_\theta \left( x_i \right), f_\theta \left( x_i' \right) \right) / \lambda \right] / B,$$

where $B$ is the batch size. We name this algorithm as TRADES-10, for it uses 10 iterations to update the adversary.

Following the notation used in previous section, we denote $f_0$ as the first layer of the neural network and $g_{\tilde{\theta}}$ denotes the network without the first layer. The whole network can be formulated as the compostion of the two parts, *i.e.* $f_\theta = g_{\tilde{\theta}} \circ f_0$. To apply our gradient based YOPO method to TRADES, following Section 2, we decouple the adversarial calculation and network updating as shown in Algorithm 1. Projection operation is omitted. Notice that in Section.2 we take advantage every intermediate perturbation $\eta^j, j = 1, \cdots, m - 1$ to update network weights while here we only use the final perturbation $\eta = \eta^m$ to compute the final loss term. In practice, this accumulation of gradient doesn't helps. For TRADES-YOPO, acceleration of YOPO is brought by decoupling the adversarial calculation with the gradient back propagation.

---

**Algorithm 1** TRADES-YOPO-$m$-$n$

---

Randomly initialize the network parameters or using a pre-trained network.
**repeat**
    Randomly select a mini-batch $\mathcal{B} = \{(x_1, y_1), \cdots, (x_B, y_B)\}$ from training set.
    Initialize $\eta_i^{1,0}, i = 1, 2, \cdots, B$ by sampling from a uniform distribution between [$-\epsilon, \epsilon$]
    **for** $j = 1$ to $m$ **do**
        $p_i = \nabla_{g_{\tilde{\theta}}} \left( \mathcal{L} \left( g_{\tilde{\theta}} \left( f_0 \left( x_i + \eta_i^{j,0}, \theta_0 \right) \right), g_{\tilde{\theta}} \left( f_0 \left( x_i, \theta_0 \right) \right) \right) \right) \cdot \nabla_{f_0} \left( g_{\tilde{\theta}} ( f_0 ( x_i + \eta_i^{j,0}, \theta_0 ) ) \right),$
        $i = 1, 2, \cdots, B$
        **for** $s = 0$ to $n - 1$ **do**
            $\eta_i^{j,s+1} \leftarrow \eta_i^{j,s} + \alpha_1 \cdot p_i \cdot \nabla_\eta f_0(x_i + \eta_i^{j,s}, \theta_0), i = 1, 2, \cdots, B$
        **end for**
        $\eta_i^{j+1,0} = \eta_i^{j,n}, i = 1, 2, \cdots, B$
    **end for**
    $\theta \leftarrow \theta - \alpha_2 \sum_{i=1}^{B} \nabla_\theta \left[ \ell \left( f_\theta \left( x_i \right), y_i \right) + \mathcal{L} \left( f_\theta \left( x_i \right), f_\theta \left( x_i + \eta_i^{m,n} \right) \right) / \lambda \right] / B.$
**until** Convergence

---

We name this algorithm as TRADES-YOPO-$m$-$n$. With less than half time of TRADES-10, TRADES-YOPO-3-4 achieves even better result than its baseline. Quantitative results is demonstrated in Table 3. The mini-batch size is 256. All the experiments run for 105 epochs and the learning rate set to $2e - 1$ initially, and is lowered by 10 times at epoch 70, 90 and 100. The weight decay coefficient is $5e - 4$ and momentum coefficient is 0.9. We also test our model's robustness under CW attack [**?** ] with $c = 5e2$ and $lr = 5e - 4$. Experiments are taken on idle NVIDIA Tesla P100 GPUs.

| Training Methods | Clean Data | PGD-20 Attack | CW Attack | Training Time (mins) |
|---|---|---|---|---|
| TRADES-10[42] | 86.14% | 44.50% | 58.40% | 633 |
| TRADES-YOPO-3-4 (Ours) | 87.82% | 46.13% | 59.48% | 259 |
| TRADES-YOPO-2-5 (Ours) | 88.15% | 42.48% | 59.25% | 218 |

Table 3: Results of "TRADES" training with PreAct-Res18 for CIFAR10