[Reviews · NeurIPS 2019]

Reviewer 1



Summary: This paper proposes a method for speeding up adversarial training by reducing the number of full forward- and backward passes during the inner loop where the adversarial examples are computed. To achieve that, the inner loop that calculates the adversarial perturbation is split into an inner and outer loop. The gradients for all but the first layer are only infrequently calculated in the outer loop, while the gradients of the first layer w.r.t to the adversarial perturbation are iteratively calculated in the inner loop. This method is motivated by casting adversarial training as time-differential game and analyzing the game using Pontryagin's Maximum Principle (PMP). This analysis indicates that only the first layer is coupled with the adversary update. The authors relate their work to "Free-m" [1], another recent work that aims to speed up adversarial training and mention that their method is a generalization with a minor change of the work in [1]. Analysis of deep learning training as control problem was previously been done in "Maximum Principle Based Algorithms for Deep Learning" [2] Originality: - The main contribution of this paper is to speed up adversarial training where the method used is motivated by casting adversarial training in the light of control theory. There has been research on the control theory perspective on training neural networks, but not related to adversarial training. Quality: - In the Algorithm following Line 138 and following Line 143 the gradients are not normalized and also not projected back on the epsilon-ball as typically done in PGD [3]. Is there any reason for that? - A non-mathematical intuition about why the adversarial update is only coupled with the first layer is not given. This would be especially interesting, as this claim is counterintuitive compared to previous methods for generating adversarials that need to differentiate through the whole network. - The experimental section lacks evaluation of stronger versions of the PGD attack (restarts, higher number of iterations) and also evaluation with non-gradient based methods. - In Line 155-159 the authors write "we take full advantage of every forward and backward propagation, […], the intermediate perturbations are not wasted like PGD-r […] which potentially drives YOPO to converge faster in terms of the number of epochs". This claim has no support in the experimental section, e.g. a plot loss over epochs. Clarity: - Theorem 1 is very hard to follow and unclear to a reader not familiar with the Maximum Principle. ○ Why is the Hamiltonian defined the way it is? - Multiple variables are not described and unclear ○ Is p in the Hamiltonian the same as the slack variable p? ○ Notation of p_i_t (used in Theorem 1) is not introduced. - Algorithm listings after Line 138 and Line 143 are missing captions - Line 157: "allows us to perform multiple updates per iteration" What iteration is meant here? Formula 144 suggests that the weights are only updated once after the m-n loop has finished. - If the coupling is only between the first layer and adversarial perturbation, why does p needs to be updated in the outer loop? - What epsilon is used during training? Significance: - Given that adversarial robustness is only slowly moving towards bigger datasets like ImageNet, the significance of a method to speed up adversarial training is high, assuming that the robustness claims are true and hold in the setting of larger datasets. - Only one other method [1] has addressed the topic of speeding up adversarial training, thus this paper is a very welcome contribution to the field. According to Table 1, the method proposed in this work further improves the training time over [1]. [1] Shafahi et al. - Adversarial Training for Free!, https://arxiv.org/abs/1904.12843 [2] Li et al. - Maximum Principle Based Algorithms for Deep Learning, https://arxiv.org/abs/1710.09513 [3] Madry et al., Towards Deep Learning Models Resistant to Adversarial Attacks, https://arxiv.org/abs/1706.06083

Reviewer 2



From what I can tell, the observation that the adversarial perturbation is coupled with only the first layer and the exploitation of this to create the YOLO method is novel and an interesting contribution to the literature that could potentially inspire a lot of follow-up work. Theorem 1 requires that f is twice continuously differentiable, but in the experiments the authors use ResNets, which have ReLU activation functions. How do the discontinuities in ReLU affect the theory? A more extensive and detailed experiments section in the main text would be good. Some additional comments/questions: -- It'd be grammatically more correct to call it "You Propagate Only Once" :) -- Shouldn't Theorem 2 be a Lemma? -- In the description of PGD, it should be eta^r, not eta^m (last line). -- Figures 3a and b are hard to see, they should be made bigger. -- Consider using a log scale for the y-axis in Figure 3a. -- It's generally better to indicate if the percentages given in tables are test accuracy or error, even if it appears obvious. -- There are a number of typos in the manuscript, for instance: "are the network parameters (line 28); "exploits"(caption of Figure 1); "min-max" (l. 66); "needs" (l. 67); "adversarial" (l. 87); "Contributions" (l. 88); rewrite the sentence starting on line 109, for instance "... is a game where one player tries to maximize, the other to minimize, a payoff functional"; "could significantly" (l. 159); "on the Hamiltonian" (l. 204); "Small CNN" (caption of Figure 3a). -- There's an unresolved reference on line 29 in the supplementary material.

Reviewer 3



This paper proposes a new algorithm for computing PGD attack, and applies the proposed algorithm to adversarial training. The experiments show that the computational cost is reduced by 2/3, and achieves the similar performance. The key insight of this paper comes from a dynamical system view of back propagation algorithm as a recurrent network. This insight, however, actually dates back to 1980's. For example, Le Cun 1988 has discussed this view in Section 3.1 http://yann.lecun.com/exdb/publis/pdf/lecun-88.pdf Moreover, the proposed algorithm is not the only way to approximate the backpropogation algorithm. There have been some truncation heuristics in existing literature, e.g., https://arxiv.org/abs/1705.08209. https://arxiv.org/abs/1807.03396 Williams and Zipser (1992), Gradient-Based Learning Algorithms for Recurrent Networks and Their Computation Complexity. Williams and Peng (1990), An Efficien Gradient-Based Algorithm for On-Line Training of Recurrent Network Trajectories, These truncation heuristics can reduce the computational cost of computing adversarial attacks. The authors did not provide any experiments to justify their method is better than these heuristics. The application to adversarial training is somehow a novel selling point, since there are not many existing works discussing the importance of truncation in adversarial training. The authors claim that the training time is reduced by 2/3~4/5. However, this highly depends on the implementation and hidden factors, e.g., library used. All experiments are based on one single realization. I did not see any details on multiple realizations and standard errors. Therefore, more experiments are needed.

[Author Response · NeurIPS 2019]

We thank the reviewers for their constructive feedback. We will incorporate these comments in the final version, and address the concerns as follows.

**General Comments: Regarding Experiment.** We put more experiments details in supplementary materials which includes the choice of $\epsilon$ mentioned by reviewer#1. We also used normalized gradient and $\epsilon$-ball projection and we'll mention this in our next version. We would also like to thank reviewer#1 and #2 for their helpful advice about writing and typesetting, which will be properly dealt with in our next version.

**Reviewer#1**

**Regarding Stronger Attack.** We conduct experiments on stronger attack, the results show our approach can defense stronger attack. The results of PreAct-Res18 on CIFAR10 are shown as follows (average of three experiments)

|  | Clean | PGD-20 | PGD-100 | PGD-1000 | CW attack |
|---|---|---|---|---|---|
| Madry | 84.89±0.19 | 42.32±0.29 | 42.13±0.27 | 41.42±0.20 | 59.30±0.16 |
| YOPO-5-3 | 83.51±0.22 | 43.94±0.20 | 43.17±0.17 | 42.52±0.36 | 60.18±0.38 |

**Regarding Clarity.** Thanks for pointing this out. The variable $p$ is a "dual" variable. Thus in Theorem 1, we need to construct a $p$ to satisfy the dual certificates. The algorithm uses an iterative scheme to find it. The variable $p$ in the Hamiltonian is the same as the slack variable $p$. The definition of Hamiltonian is brought from physic and is well known in the control community. It can also be understood as a Fréchet Dual of the original problem.

**Regarding Free-m.** We would like to point out that the Free-m method is an independent and *concurrent* work (was put on arXiv on April 30 which was just before the NeurIPS' deadline). In our paper, we also show that their method is a special case of ours, namely YOPO-$m$-1. The epsilon used (1-7) in Free-m paper for imagenet is wired (too small) and the accuracy is far from the state-of-the-art report [1]. Imagenet is still a hard problem mainly due to limited computation resources, and we are still working on it. ([1] Feature Denoising for Improving Adversarial Robustness arXiv:1812.03411)

**Regarding using the first k-layers be used for the inner-loop adversary.** It is flexible to try $k$ other than 1, but in our experiment, selecting $k = 1$ works the best. We will include an ablation study in the final version.

**Regrading the analysis of $m$ and $n$.** Thanks for your suggestion and we will add more ablation study over this. The analysis could be found in Line145-154, we also use YOPO-3-5 and YOPO-5-3 to empirically justify the analysis.

**Reviewer # 2**

**Regarding Twice Continuously Differentiability.** The set of non-differentiable part of ReLU is of measure zero. Thus we do not think this will affect the algorithm a lot. The BP algorithm typically requires the activation function to be differentiable but works well empirically. Reviewers can consider there exist a really good smooth function to approximate ReLU. *First order differentiability is enough for the theory in our paper, while twice continuous differentiability may be required for further convergence analysis.*

**Regarding Theorem 2.** Theorem 2 is used to show the relationship between our algorithm and PMP, and is important for that matter.

**Reviewer#3**

**Regarding comparison with previous work.** First of all, as reviewer#1 mentioned, one of the main contributions is discovering the benefits of the control perspective in the *adversarial setting*. We agree the control perspective is not a new idea in deep learning and we have already cited the original Lecun's BP paper and other related papers. At the same time, the long training time is the *main* issue when scaling adversarial training to a larger dataset and networks. That's why most of the adversarial training papers just test CIFAR10. In our work, we showed the power of control perspective in accelerating the heavy training procedure, which we think will help the community to scale up their experiment.

Secondly, there seems to be some misunderstanding that our work is using control to model **feed forward network** but not **RNN**. It's **not time-homogeneous**. It is not clear to use how the BPTT algorithms could be applied in our setting.

Finally, our splitting method provides a new perspective on solving the optimality condition. This new perspective not only provides a description of the algorithm in a more general setting, but also inspires algorithms beyond back-propagation based training.

**Regarding the training time.** First of all, the computational cost (e.g. FLOP) of YOPO is theoretically smaller than the original adversarial training, typically 1/5-1/4 times smaller. The code is provided in the supplementary for reproducibility. All codes are written in Pytorch. There is also an unofficial TensorFlow code on Github showing that YOPO is quite efficient.

[Meta-Review · NeurIPS 2019]

This paper is somewhat borderline: it contributes some novel ideas from control theory to adversarial learning on one hand, yet also leaves open questions and a somewhat weak experimental evaluation. In the end, the positive aspects slightly outweigh the concerns.